# Using Infrared Imagery to Assess Fire Behaviour in a Mulched Fuel Bed in Black Spruce Forests

**Brett Moore** [1,2,*] , **Dan K. Thompson** [1] , **Dave Schroeder** [2] , **Joshua M. Johnston** [3] **and Steven Hvenegaard** [4]

1   Canadian Forest Service, Natural Resources Canada, Northern Forestry Centre, Edmonton, AB T6H 3S5, Canada; Daniel.Thompson@canada.ca
2   Alberta Agriculture and Forestry, Wildfire Management Branch, Government of Alberta, Edmonton, AB T5K 1E4, Canada; Dave.Schroeder@gov.ab.ca
3   Canadian Forest Service, Natural Resources Canada, Great Lakes Forestry Centre, Sault Ste Marie, ON P6A 2E4, Canada; Joshua.Johnston@canada.ca
4   FPInnovations, Wildfire Operations Research Group, Edmonton, AB T5G 0X5, Canada; Steven.Hvenegaard@fpinnovations.ca
*   Correspondence: Brett.Moore@canada.ca

**Abstract:** An experimental fire was conducted in one-year-old mulched (masticated) boreal fuels, where all aboveground biomass was mulched with no stems removed or left standing. Typical mulching practices remove remnant biomass; leaving biomass in situ reduces overall management input. While fuel quantities were not explicitly reduced, availability of fuels to fire was reduced. Infrared imagery was obtained to quantify rate of spread and intensity to a 1 m resolution. In-stand totalizing heat flux sensors allowed for the observation of energy release near the surface. When compared with the pre-treatment fuel-type M-2 (mixedwood, 50% conifer), rates of spread were reduced 87% from an expected 8 m min$^{-1}$ to observed values 1.2 m min$^{-1}$. Intensity was also reduced from 5000 kWm$^{-1}$ to 650kWm$^{-1}$ on average. Intermittent gusts caused surges of fire intensity upwards of 5000 kW m$^{-1}$ as captured by the infrared imagery. With reference to a logging slash fuel type, observed spread rates declined by 87% and intensity 98%. Independent observations of energy release rates from the radiometers showed similar declines. As mulching is a prevalent fuel management technique in Alberta, Canada, future studies will contribute to the development of a fire behaviour prediction model.

**Keywords:** prescribed fire; mulch; mastication; boreal; black spruce

## 1. Introduction

Fuel management to reduce fire intensity, in particular mulching (also known as mastication) [1], has become increasingly prevalent as communities seek for ways to prepare for and defend from wildfires. Recent prolonged fire seasons with multiple Wildland-Urban Interface (WUI) fire events [2] have resulted in an increased focus on programs aimed at community protection and preparedness; in Canada, this is primarily achieved through the FireSmart program [3]. Fuel management has been a priority area for many wildfire management agencies, as fuels are the only component of the fire environment that managers have direct control over. A number of fuel management techniques are being used on the landscape that are similar to the mulching performed in this study, such as stand thinning and clearcutting. During this mulching treatment, remnant fuels remain in situ as opposed to a typical pile and burn strategy to remove the fuels. This is a key difference from standard fuel management techniques as the total available fuel is not reduced but converted to mulch and displaced

to the surface fuel layer. Immediate fuel availability is altered as there is no longer a forest crown; however, bole wood is more accessible to fire than it would be in an intact stand. These departures from typical fuel management practices yield an interesting opportunity to assess the response of an increasingly common fuel treatment to fire. An experimental fire in a partially mulched black spruce stand (following an alternating strip pattern) showed minimal reduction in fire intensity due to the fire readily breaching the 4–6 m wide mulch gap between trees [4]. Fire behaviour in stands with mastication of only sub-canopy trees or shrubs has been documented [5]. Fire behaviour in completely mulched (with no stem removal or retention of standing trees) in low-volume boreal forest is unknown, despite the treatment being common along wide utility rights-of-way and at the edge of industrial sites in boreal western Canada.

The Canadian Forest Fire Danger Rating System (CFFDRS) contains two subsystems for defining wildfire hazard, the Canadian Forest Fire Weather Index System (FWI) and the Canadian Forest Fire Behaviour Prediction System (FBP) [6]. The FBP system is used to predict the potential spread and intensity of fire based on 17 standard fire behaviour models. There are 17 fire behaviour models in five broad categories: C—coniferous fuels (7 models), D—deciduous fuels (1 model), M—mixedwood fuels (4 models), O—open (grass) fuels (2 models), and S—slash fuels (3 models). Mulch fuels do not directly correspond to any of the 17 standard fuels; there is a need to gather field observations of fire behaviour in this novel fuel type. Experimental fires are required to provide information toward the development of a mulch empirical fire spread model. Mulching is expected to reduce the overall rate of spread and intensity of fire in these fuels; however, the magnitude of that reduction is difficult to assess without experimental case studies as this treatment does not remove fuels. Additionally, the intensity of the mulching influences fuel distribution, which has subsequent effects on fire intensity.

Nearby instances of rapid wildfire spread in recently-burned forest that has traditionally been considered a barrier to spread [7] increases the importance of a detailed understanding of fuel treatments if they are to be implemented near communities. The objective of the experimental fire was to better understand fire behaviour in open mulch fuels, with a focus on rate of spread and head fire intensity (hereafter intensity). We also assess the assumption that the slash fire behaviour model (dominated by post-harvest fine branchwood and dead foliage) from the Canadian Forest Fire Behaviour Prediction System (FBP) is an appropriate surrogate. Additionally, we compare the results from this experimental fire with the results from an adjacent plot that was burned as an experimental crown fire in May 2019 [8]. In this case study, we present a single ignition and spread experiment over space and time using airborne infrared to provide an insight not only to the mean rate of spread, but also the variability within an event. The case study is presented as a template for gathering further data in the fuel type to eventually create an empirical rate of spread model for the fuel type.

## 2. Materials and Methods

### 2.1. Site Description

This experimental fire took place at the Alberta Agriculture and Forestry's Pelican Mountain FireSmart Fuel Management Research Area south of Wabasca, Alberta, located at 113.57° W, 55.71° N (Figure 1). While a number of units are visible within Figure 1 the key units are Unit 1 and Unit 5, which were both lowland boreal spruce sites with mulching activity that were burned 1 year after treatment. The site is intended to provide a platform for researchers to perform a number of different studies in a variety of fuel complexes. The fire took place on the east side of the mulched containment perimeter, on 31 May 2017 (Figure 2). The area is a typical black spruce (*Picea mariana*)/jack pine (*Pinus banksiana*) complex, with lower elevations dominated by black spruce, while higher sites have an abundance of jack pine. The highest elevations are dominated by trembling aspen (*Populus tremuloides*). The broader mulched area has elevations ranging from 634 m to 640 m, with an average elevation of 637 m. The area burned has elevations ranging from 636 m to 638 m, with an average elevation of 638 m. While elevation is an important factor when calculating fire behaviour potential, the variability

of elevation on this site is considered flat. Soils underlying the mulch have an organic layer 10–15 cm at higher elevations, and 40 or more cm of organic soil at the north and western edges of the burn plot. Clay mineral soils are found throughout the site.

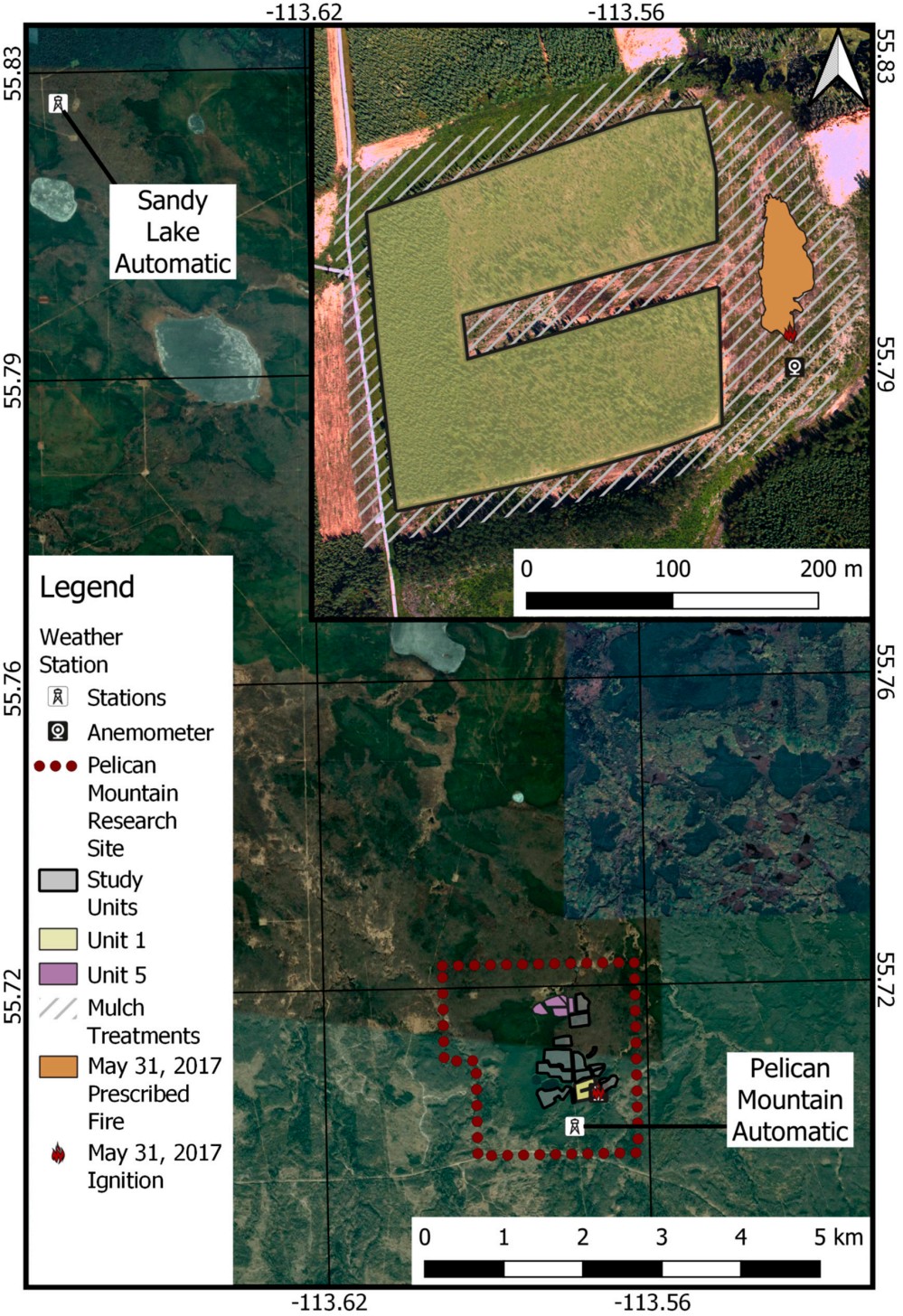

**Figure 1.** Pelican Mountain Research Site Map—broader pelican mountain research area and local weather stations. Inset—May 31 mulch prescribed fire perimeter to the east of experiment plot Unit 1. The experimental crown fire in Unit 5 conducted in 2019 is described in [8].

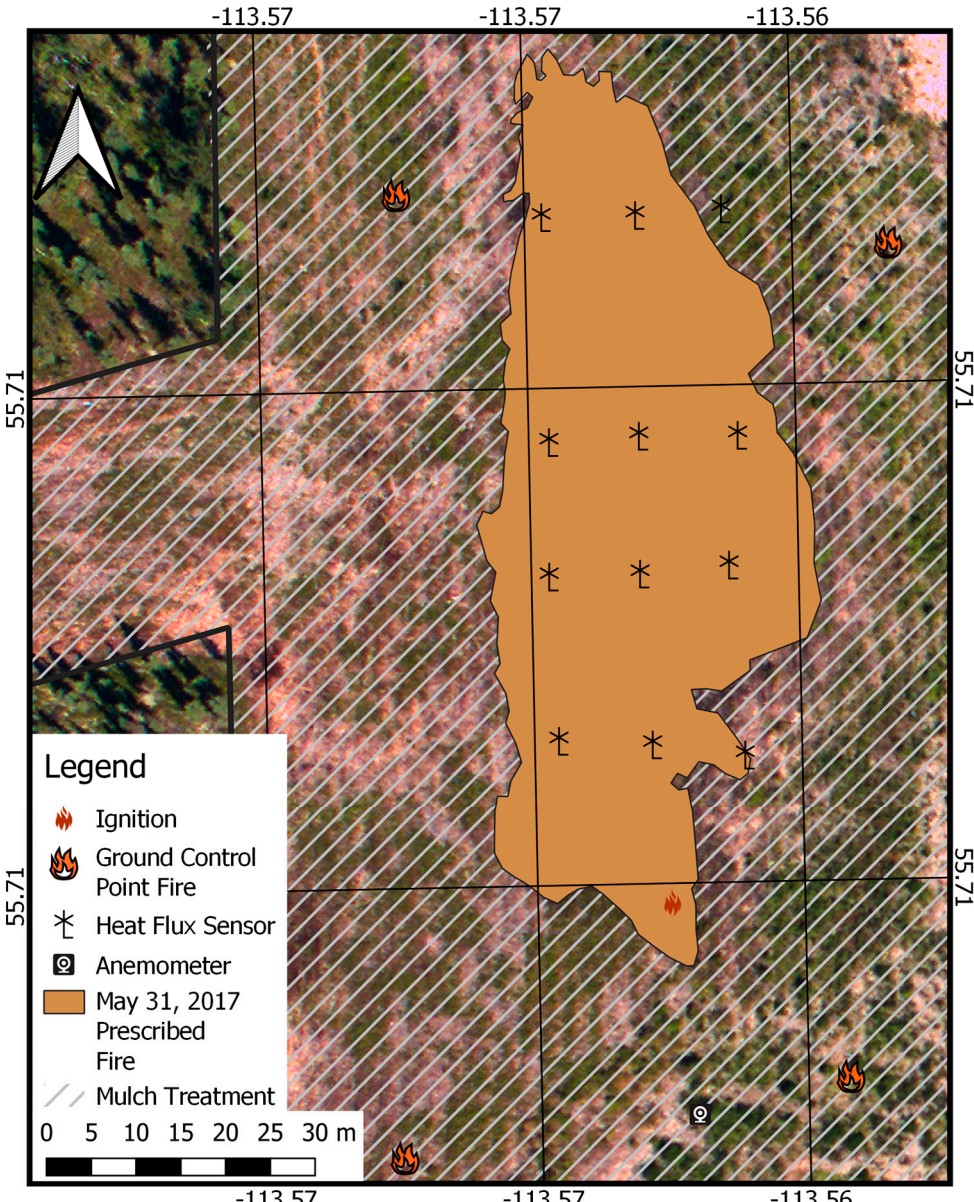

**Figure 2.** Overview of the experimental design, including heat flux sensor placement, ground control point fires, and the on-site anemometer location.

The mulching treatment was performed in the winter one year prior to the burn. The time lag post-treatment was sufficient for mulched fuels to fully dry while minimizing decomposition common in older treatments. Prior to mulching, the fuel type in the area was described as a 50% mixedwood of aspen and black spruce, or M2-50% as per the Canadian Forest Fire Behaviour Prediction system. The mulching protocol in the area was similar to the mulch protocol used in other studies at Pelican Mountain. Specifically, a GyroTrac G-T 25XP with the TOMA-AX 700HF cutter head, a 2.2 m wide mulching tool, passed over the site twice. Previous studies in the area have described this treatment as a "normal" intensity of mulching, by which "stems are knocked down and processed with at least two passes, yielding a uniform fuel bed with a minimal amount of intact round wood" [9]. The distribution of mulch on the site can be categorized into 4 main size classes (SC) based on diameter: SC1 (0–0.5 cm), SC2 (0.5–1 cm), SC3 (1–3 cm), and SC4 (3–5 cm). The mulch particle composition is assumed to follow the distribution measured in [9] at other mulched areas at the same site and equipment: SC1 58%, SC2 17%, SC3 20%, and SC4 5%. Average pre-burn mulch thickness in the plot was 9.5 cm, and varied between 5 and 17cm. The average mulch fuel load was 8.3 kg m$^{-2}$, and varied with depth between

3 and 16 kg m$^{-2}$. Pre-mulch fuel load was not collected. Along the eastern edge of the fire area, an aspen-dominated mixedwood stand provided a barrier to prevent the fire from escaping. Forests in the region typically have a fire return interval of 60 to 80 years [10] with increasing fire frequency in recent decades [11].

## 2.2. Weather

Climatological data was collected from Sandy Lake tower (13.5 km to the NW), which has been seasonally (April–October) active since 1977. The temperature and relative humidity during the fire was beyond the 30 year 90th percentile for the area (Table 1). During the month of May in 2017, 20 of 31 days were below normal precipitation (<1 mm daily), though the monthly precipitation was average (47 mm). From May 25 to May 31, no precipitation was recorded at the research site. In this absence of precipitation, the mulch fuels dried to a state that would permit combustion in the uppermost 5 cm layer of fuel [12]. Complete snow loss was observed in February in open mulched area, suggesting early snow loss and fuel drying. This would affect coarse woody debris moisture during melt, as the snow depth before thaw would be lower than normal. On the day of the burn, fire weather indices indicated that coarse woody debris would be readily available for combustion (Table 2).

**Table 1.** May 31 temperatures (on site sonic anemometer and Pelican Mountain temporary weather station) and 30-year average (Sandy Lake Permanent weather station). Values followed by percentile from Sandy Lake 1987–2016 data in the month of May.

| Weather Variable | May 31—Anemometer | May 31—Temporary Automatic | Sandy Lake 30-Year Average |
|---|---|---|---|
| Temperature (°C) [*n*th percentile] | 28 (99) | 33 (100) | 14 |
| Relative Humidity (%) [*n*th percentile] | 23 (97) | 37 (63) | 47 |
| Wind Speed (km h$^{-1}$) [*n*th percentile] | 10 (47) | 7 (25) | 12 |

**Table 2.** Pelican Mountain Temporary Automatic Fire Weather Indexes from May 31. The Canadian Fire Weather Index system codes given: FFMC—Fine Fuel Moisture Code, DMC—Duff Moisture Code, BUI—Build up Index, DC—Drought Code, ISI—Initial Spread Index, FWI—Fire Weather Index.

| | | Fire Weather | | | |
|---|---|---|---|---|---|
| FFMC | DMC | BUI | DC | ISI | FWI |
| 93 | 63 | 69 | 175 | 9 | 24 |

The Pelican Mountain temporary automatic weather station (roughly 1 km away) achieved a maximum temperature of 33 °C on May 31. The sonic anemometer on site achieved a maximum temperature of 28 °C with a low relative humidity of 24%. Temperature and humidity far exceeded normal values of Sandy Lake's 30-year noon fire weather observations temperature record (18 °C) for May 31.

## 2.3. Prescribed Fire and Data Collection

The primary data for this case study is fire rate of spread; data were captured with a handheld infrared camera operated from a helicopter that was in a stationary hover approximately 300 m above ground level. Due to the nature of infrared imagery taken from a helicopter, georectification was required. Georectification was achieved with four ground control point fires that were lit at least 10 m away from the potential spread of the prescribed fire and were maintained for the duration of the experiment. Assessment of rate of spread from georectified imagery was completed using the time at which temperature surpassed a threshold of 753 K from a calculation of fire arrival time in seconds [13]. This arrival time grid allowed for the calculation of rate of spread across the burn area using the rate

of change across arrival time (Figure 3A). Rate of change, or slope, across the arrival time grid was calculated using the terrain function within the Raster package in R using an eight neighbour search window [14]. Rate of spread (RoS) is then calculated with Equation (1), where the slope is calculated from the arrival time in minutes and the resolution is the resolution of the gridded arrival time data.

$$RoS = \frac{1}{\left(\frac{\tan(slope)}{resolution}\right)} \tag{1}$$

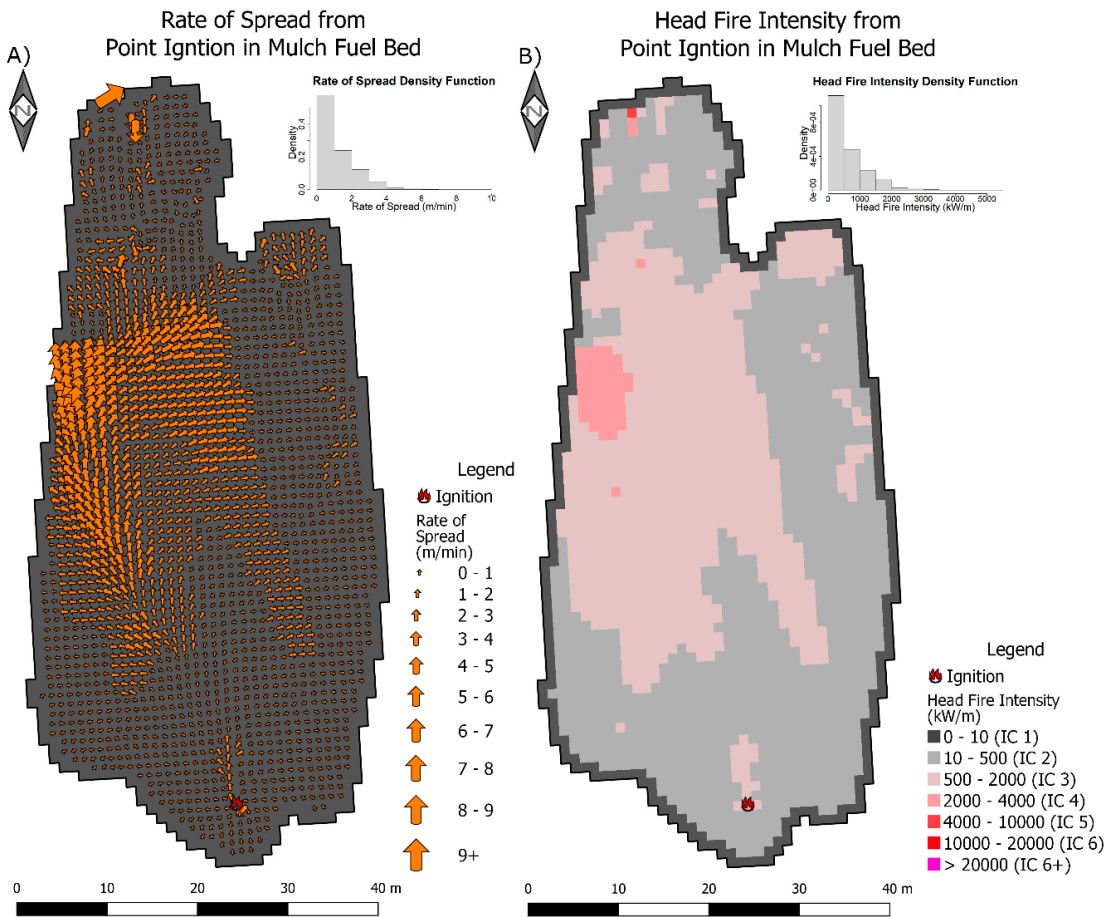

**Figure 3.** Maps of (**A**) rate of spread (**B**) head fire intensity with intensity class (IC). Head fire intensity used the average observed fuel consumption of 1.8 kg m$^{-2}$.

In order to capture weather information, an on-site sonic anemometer (ATMOS-41) at 1.4 m above ground level recorded wind speeds, temperature, and humidity from 25m upwind of the ignition point every 10 s. Twelve in-fire forward-facing totalizing heat flux sensors [15] recorded total heat (radiative plus convective) flux 30 cm above the fuel bed (Figure 2). These sensors provide a relative heat flux in MJ m$^{-2}$ as well as peak energy inputs in kW m$^{-2}$, but are not linearly relatable to Byram's fire line intensity. Depth of burn was measured using burn pins co-located with heat flux sensors.

## 3. Results

### 3.1. Fire Observations

The fire was started from a point ignition at 15:00 local time. Wind direction as recorded near the ignition point by a sonic anemometer showed consistent SE winds with brief gusts to the SW (Figure 3). The average wind speed during the duration of the burn was 11.3 km h$^{-1}$ (extrapolated

to 10-m height wind speed equivalent from the 1.4 m observation height). Maximum recorded wind gusts at 2 s intervals approached 40 km h$^{-1}$, though 90% of wind observations fell below 20 km h$^{-1}$. Fire rate of spread ranged from 0.09 m min$^{-1}$ to 10 m min$^{-1}$. Overall, the fire's average forward rate of spread was 0.6% that of the average wind speed. The mode and average are 0.19 and 1.18 m min$^{-1}$, respectively, and have a strong right skew as 80% of the data are below 1.9 m min$^{-1}$ (Figure 3). Average depth of burn (assessed immediately after the fire reached its further extent) in the mulch fuel was 3.5 cm, and was as high as 6.5 cm. Two of 16 depth of burn pins recorded values no different from zero. In total, only 2.5% of the fire growth area was intensity class (an ordinal ranking of fire intensity at critical suppression thresholds [6]) of 4, or 2000 kW m$^{-1}$ and above, providing a challenge to ground crews with power pumps and ample water, who tested suppression ability at the fire's edge (Figure 4). The intensity was calculated by using 50% of the plot-averaged total post fire fuel consumption (1.8 kg m$^{-2}$), a fraction derived from laboratory observations of mulch burning kinetics in the flaming phase [16]. Consequently, the remaining 50% fuel consumption is assumed to have occurred after the fire front passed; head fire intensity calculations only consider fire frontal passage. Intensity ranges from roughly 50 to 5000 kW m$^{-1}$, spanning operational intensity classes 1 through 5 (Figure 4b) [6]. Heat flux as measured by in-fire sensors ranged from 0.1 to 1.1 MJm$^{-2}$ with an average heat flux of 0.4 MJ m$^{-2}$. The average totalized heat flux of 0.4 MJ m$^{-2}$ is only 10% of that of a continuous crown fire conducted at the same site in 2019, and the maximum observations of 1.1 MJ m$^{-2}$ are comparable to the heat flux of a thinned conifer stand with intermittent torching [8].

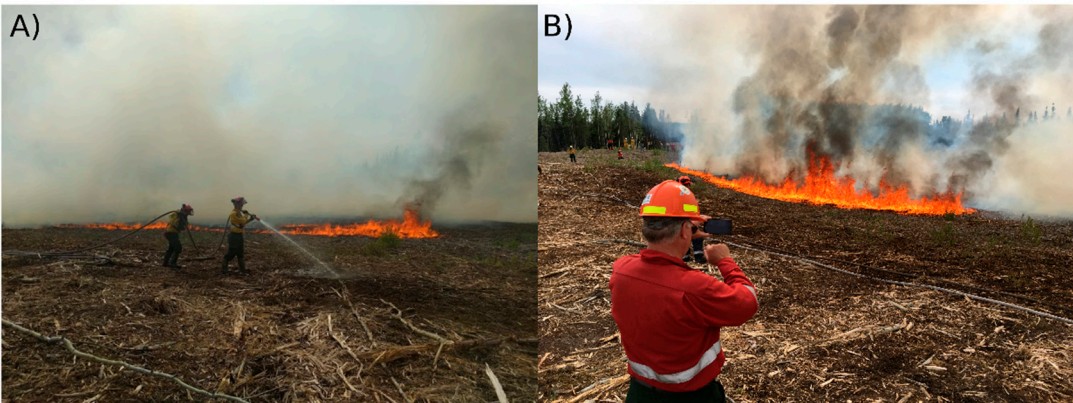

**Figure 4.** (**A**) Typical mulch fire behaviour dominated by low flames less than 0.5 m in height. (**B**) High intensity fire behaviour (flames up to 2 m in length) observed intermittently during when winds approached 20 km h$^{-1}$.

*3.2. Comparison to Fire Behaviour Models*

Management agencies use the Field Guide to the Canadian Forest Fire Behaviour Prediction System to associate intensity class 5 (4000–10,000 kW m$^{-1}$) with a change in wildfire suppression tactics from direct (head firefighting) to indirect (flank and back firefighting) attack [6]. While the intensity is manageable by suppression crews, this fuel shows the capacity to have very high head fire intensities for portions of the fire head (Figure 4). Potential fire behaviour for a fire of this nature would have been assessed using a surrogate fire behaviour model, in this case S-2. Notably, S-2 has a considerable head fire intensity of 28,000 kW m$^{-1}$, yet the observed head fire intensity was 650 kW m$^{-1}$ (Table 3).

**Table 3.** Canadian Forest Fire Behaviour Prediction System values for observed mulch fire behavior and comparison fuel types S-1 Jack or Lodgepole Pine Slash, S-2 White Spruce/Balsam Slash, S-3 Coastal Cedar/Hemlock/Douglas-fir Slash, and M-2 Boreal Mixedwood Green (50% Conifer).

| | Fire Behaviour | | |
| --- | --- | --- | --- |
| Fuel Type | Rate of Spread (m min$^{-1}$) | Head Fire Intensity (kW m$^{-1}$) | Fuel Consumption (kg m$^{-2}$) |
| Observed | | | |
| Mulch | 1.2 | 650 | 1.8 |
| Reference Model | | | |
| S-1 | 16 | 35,000 | 7 |
| S-2 | 8 | 28,000 | 12 |
| S-3 | 13 | 94,000 | 23 |
| M-2 (50% conifer) | 8 | 5000 | 2.8 |

## 4. Discussion

The observed experimental mulch fire was considerably less intense and slower-spreading than any of the slash benchmark fire behaviour models (Table 3). Fire in mulch spread 88% slower than slash (S-2) and was 98% less intense. This demonstrates the need for additional data in mulch to define a unique fire spread model. Observations of fireline intensity that decline due to mulching are broadly similar to masticated fuels in the U.S. [5], with notably thicker fuel beds, higher wind speeds, and consequently somewhat higher flame lengths in this study.

While the fuels superficially visually resemble slash, the experimental fire burned at far lower rates (forward rate of spread 0.6% of wind speed vs. 4% of wind speed for a spruce (S-2) slash model) model in the Canadian Forest Fire Behaviour Prediction (FBP) system. A key physical fuel bed contrast between mulch and slash fuels is the presence of coarse fragments derived from stem wood present in mulch; similarly, the fraction of dead and cured conifer foliage is very low in mulch as compared to slash [9]. As a result, the fraction of very fine fuels (i.e., foliage) is much higher in slash than mulch, and is likely a contributor to the lower fire intensities observed here. The deposition of mulch is also different from slash; mulch is intended to be a uniformly distributed material, while slash is remnant material from a harvesting operation [6]. The results from the burn follow the findings of Rothermel in his initial description of the United States fire behaviour models where high density reduces combustion rates [17]. Further, the Rothermel approach helps to describe the high intensities seen with low rates of spread due to a change in porosity of the fuel structure with no change in the total fuel load [17], a feature not present in Canadian fire spread models of fixed fuel density.

To compare the expected fire behaviour in the pre-treatment stand, fire behaviour potential was assessed with the standard mixedwood fuel type M-2 (50). A predicted pre-treatment modelled spread rate of 8 m min$^{-1}$ far exceeds observed values of 1.2 m min$^{-1}$. There was also a marked reduction in intensity, from an expected 5000 kW m$^{-1}$ to the observed 650 kW m$^{-1}$. The low average intensity was interrupted by brief surges of higher spread rate (Figure 4) and consequential peaks of intensity as high as 5000 kW m$^{-1}$. While the M-2 (50) fuel type serves as a surrogate in the absence of quantified fuel loading, future mulching studies would benefit greatly from measured pre-treatment fuel loading. Quantifying the available fuel would allow for a better estimation of the fuel availability reduction than mulching without physically removing material yields. We can assert that a reduction in available fuel has occurred due to the reduction in expected and observed fire behaviour.

In the Canadian boreal forest, a fire weather index (FWI) of 19 is a commonly used benchmark for a 50% probability of large wildfire spread potential [18]. May 31 had an FWI of 25, which is categorized as very high FWI within Alberta. Fire whirls formed at times and dissipated rapidly, which is indicative of brief periods of extreme fire behaviour. During the fire whirl events, spotting was minimal, and firebrands were not present at the rates anticipated by individuals participating in the experiment. During the experiment on May 31, previously burned area from the day prior was reignited by firebrands. Reignition potential in burned fuel treatments will need to be further explored

when considering the management implication of mulching fuels around communities. A repeat burn schedule may be necessary to achieve the desired hazard reduction.

FireSmart vegetation management is not intended to stop a wildfire, rather the managed stand should enhance suppression efforts [19]. The fire behaviour documented in the 100% mulch suggested a greater potential for suppression success relative to other FBP fuel types or partially mulched spruce stands [4]. The results indicate that future test burns should focus on conditions with higher wind speeds and drier conditions to determine if there is a weather or FWI threshold at which point mulch fires cannot be safely actioned by fire crews.

The use of overhead infrared scanning illustrates potential to document fire behaviour at a level of detail unavailable to researchers originally developing FBP. Whereas a single burn yielded a single data point [5], the resolution of the infrared data, and variability, indicates the potential for multiple data points. Use of these methods could reduce the number of test burns or wildfire operations needed to collect enough data points to build a fire behaviour curve.

The influence long duration fires have on mulched lowland boreal ecosystems is not yet fully understood, as such systems typically only burn off 5–10 cm of low-density surface moss [20]. Surface temperatures recorded by the overhead infrared camera in the burned area that were above 70 °C persisted for 40 min and temperatures over 100 °C persisted for 30 min. Long duration high temperatures have an opportunity to negatively affect subterranean ecosystems [21,22]; however, we did not record soil temperatures. There was also notable shrub recovery within one year (May 2018) to approximately 50% cover, primarily of the flammable ericaceous shrub *Rhododendron groenlandicum*. Rapid post-fire shrub recovery in concert with remnant mulch and organic soil itself represents a novel fuel complex that is in need of further investigation, especially if grass species that are typically problematic after timber harvest in the region [23] emerge.

## 5. Conclusions

This case study of fire behaviour in boreal mulched fuels has showed an effective reduction in spread rate and intensity during relatively high fire weather indices (hot and dry) down to levels within the realm of direct suppression. High-resolution overhead infrared imagery revealed brief periods of fire intensity above levels of safe direct attack, despite mean intensity values well within safety limits for direct attack. Experimental fires for assessing suppression capacity (e.g., [8]) should as much as possible employ overhead infrared progression mapping to capture both the mean and the variance in spread rates. Slash fire behaviour models were found to be inappropriate surrogates to boreal mulched fuels, and additional experiments are required to define a mulch spread model. Future similar fires will be key in understanding the parameters under which mulch treatments are effective. Improved knowledge of the vegetation recovery and post-fire flammability will be key in understanding the efficacy of applying low-intensity prescribed fire to boreal mulch treatments. This study and subsequent studies are intended to serve as data for the development of a mulch fuel type in black spruce dominated forests, which would be applicable across boreal North America.

**Author Contributions:** Conceptualization, B.M., D.K.T., D.S., J.M.J. and S.H.; Formal analysis, B.M., D.K.T., D.S., J.M.J. and S.H.; Funding acquisition, D.S.; Investigation, B.M., D.K.T., D.S., J.M.J. and S.H.; Methodology, B.M., D.K.T., D.S., J.M.J. and S.H.; Project administration, B.M. and D.S.; Supervision, B.M.; Visualization, B.M. and D.K.T., Writing—original draft, B.M., D.K.T., D.S., J.M.J. and S.H.; Writing—review and editing, B.M., D.K.T., D.S., J.M.J. and S.H.; All authors have read and agreed to the published version of the manuscript.

**Funding:** This research received no external funding.

**Acknowledgments:** The Alberta Wildfire Management Branch created and supported this prescribed burning through planning, logistics, and operations. The Incident Management Team, as well as all the firefighters and support staff are acknowledged for their hard work and flexibility around scientific objectives. Alberta-Pacific Forest Industries Inc. provided in-kind support via a modified harvest of the aspen stands surrounding the burn unit. Bigstone Cree First Nation provided crews to do the thinning work and essential firefighting staff and community support for the project. FPInnovations for key data collection and discussion.

**Conflicts of Interest:** The authors declare no conflict of interest.

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
