# Peer review of "Using Infrared Imagery to Assess Fire Behaviour in a Mulched Fuel Bed in Black Spruce Forests"

_fire, doi:10.3390/fire3030037_

Round 1

Reviewer 1 Report

I have written comments within the manuscript and should take a look. There are some problems in the methods that can be rectified. The overall presentation of the manuscript needs to be written more smoothly. It is almost like a series of bullet points were made into paragraphs. I know there is good material there but it does not seem to be communicated well. You should try to provide more background to the study in the introduction.

Author Response

We thank Reviewer 1 for their careful and considerate suggestions, and revise the manuscript as detailed below:

should it not be severity? or intensity?

>>> Changed as suggested to fire intensity

50% mixedwood aspen and 50% spruce? It is unclear what the fuel type is.

>>> revised to “(mixedwood, 50% conifer)”

which is????? How was it mulched?

>>> We now give more details on the machinery used in mulching: “Specifically, a GyroTrac G-T 25XP with the TOMA-AX 700HF cutter head, a 2.2 m wide mulching tool passed over the site twice. Previous studies in the area have described this treatment as a “normal” intensity of mulching, by which “stems are knocked down and processed with at least two passes yielding a uniform fuel bed with a minimal amount of intact round wood” [6]”

You did not verify this distribution by sampling? What do you have to validate this assumption?

>>> we specify this distribution is from other mulch plots on the same site using the same equipment, which we can assume follows a similar size distribution.

prevent the fire from

>>> Changed to “mixedwood stand provided a barrier to prevent the fire from escaping”

Location relative to the study site?

>>> we revised this to read “…tower (13.5 km to the NW)…”

Only one thermometer? Did you not take sample readings across the study area? Where was it located on the study site?

>>> they were anecdotal temperature measurements taken with the infrared camera prior to ignition.  We’ve removed this reference.

Burn

>>> 

do you mean figure 2a?

>>> We have fixed all figure references and altered the order in which they appear.

>>> We have added additional information on the in-fire radiometers: “In-fire forward-facing totalizing heat flux sensors (Anderson and Macdonald, 2014) recorded total heat (radiative plus convective) flux 30 cm above the fuel bed (Figure 2).  These sensors provide a relative heat flux in MJ m-2 as well as peak energy inputs in kW m-2, but are not linearly relatable to Byram’s fire line intensity.”

across the entire study area?

>>> We now clarify in the text that there were 12 heat flux sensors, and their layout geometry is given in Figure 2.

what is the primary data?

>>> we clarified this to: “The primary data for this case study is fire rate of spread data”

is this reference [11] ?

>> yes, reference has been fixed.

skew as 80%

>> revised as suggested.

The intensity was calculated by using

>>> revised as suggested to “The intensity was calculated by using 50% of the total post fire fuel”

How much is considerable?

>>> We revised this to be more specific: “

How many?

>> We’ve now included the number of radiometers and burn pins included in these values.

incomplete sentence

>>> Revised to “While the fuels vaguely resemble slash, the experimental fire did not behave like the reference slash model in the Canadian Forest Fire Behaviour Prediction (FBP) system.”

This sentence is confusing as written and needs rewritten; but I can't understand what you are trying to say to suggest a rewrite.

>>> revised to “A typical slash fuel does not have the stem wood present as is the case with mulch; similarly, the fraction of dead and cured conifer foliage is very low in much as compared to slash [5].  As a result, the fraction of very fine fuels (i.e. foliage) is much higher in slash than mulch, and is likely a contributor to the lower fire intensities observed here.”

reported fire intensity

>> revised as suggested

experiment as there

>>> revised as suggested

fire which

>> revised as suggested

study. We

>> sentence split as suggested

I did not see the mean recorded temperatures or standard deviations. How many observations? What were the exact fuel conditions at each measurement point?

>>> we revised this to: “Long duration high temperatures have an opportunity to negatively affect subterranean ecosystems, however we did not record soil temperatures [19–21].  “ and spoke in a qualitative manner about the mulch decomposition and shrub regrowth, without specific mention of soil temperature values.

Reviewer 2 Report

This study uses an experimental site and fire to better understand fire behavior in open mulch fuels. The authors focus on rate of spread and head fire intensity. They also evaluate the slash fuel model from the Canadian Fire Behavior Prediction System and compare to the experimental fire. This is an important topic, but the manuscript needs more detail before it should be published.

Generally, the abstract lacks detail about what was done in the study and concepts/topics are mentioned, but not described.

Similarly, the introduction is short and doesn’t convey the need for this case study. What other fuel management techniques have been used and evaluated? Why is this particular case study of value to the community?

The site description can be improved by mentioning the soil type and soil moisture conditions. These are valuable information for a study about fuels.

Further, a site map is needed that should include the location of the sites evaluated and the Sandy Lake tower. The site map could also be useful to show the burned area and extent, fuel composition and arragenement, and sensor locations.

Other comments:

L 11 where is Pelican Mountain research site?

L12-13 this sentence is awkward, treated how? This is also a good place to say what was done in the study.

L17 superscripts needed for units

L18-19 specifically what is expected in a logging-slash fuel type?

L37-38 more information about the fire and area are needed in Section 2.1. Further, a site map is needed that should include the location of the sites evaluated.

L55-60 should be introduced much earlier in the paper, such as in the introduction.

L59 what is Unit 5 -- what is the composition, general description of this site and why is is used for comparison?

L62 add the tower to a site map.

L64-65 the dates need a year

Section 2.2 weather discussion would benefit from # of antecedent dry days. What is the average precipitation?

Table 2 caption and table need to be improved. What is FWI, S-1, S-2, S-3, and M-2?

L99 define CFFDRS. Also need more description about the model that is used.

L102 needs a space between min-1(Figure)

L105 quantify what “considerable fuel consumption” is

Figure 2 – remove the titles at the top of the figure A and B (i.e. “rate of spread from point ignition in mulch fuel bed”) and incorporate into the Figure captions.

L116 use a superscript for units in 1.8 kg/m2

L117-118 needs more context for those unfamiliar with this model. This should be described prior to the results (in the methods section).

L117-127 is “discussion” and should be moved

L124 spell out US

L131 what dates were the visits? What were the preceding and prevailing precipitation and temperature conditions like? How much shrub recovery, where, etc.?

L150-152 show this or provide supporting evidence

L152-153, why is this important to note?

L171 what is spotting

L181 how can this case study be applied to other areas or management?

Author Response

We thank Reviewer 2 for their comments and revise the manuscript accordingly as detailed below.

Reviewer 2

Open Review

English language and style

( ) Extensive editing of English language and style required
( ) Moderate English changes required
(x) English language and style are fine/minor spell check required
( ) I don't feel qualified to judge about the English language and style

Yes

Can be improved

Must be improved

Not applicable

Does the introduction provide sufficient background and include all relevant references?

( )

( )

(x)

( )

Is the research design appropriate?

( )

(x)

( )

( )

Are the methods adequately described?

( )

( )

(x)

( )

Are the results clearly presented?

( )

(x)

( )

( )

Are the conclusions supported by the results?

( )

(x)

( )

( )

Comments and Suggestions for Authors

This study uses an experimental site and fire to better understand fire behavior in open mulch fuels. The authors focus on rate of spread and head fire intensity. They also evaluate the slash fuel model from the Canadian Fire Behavior Prediction System and compare to the experimental fire. This is an important topic, but the manuscript needs more detail before it should be published.

Generally, the abstract lacks detail about what was done in the study and concepts/topics are mentioned, but not described.

>> We have extensively revised the abstract to be more clear on what was measured, and how it was done.

Similarly, the introduction is short and doesn’t convey the need for this case study. What other fuel management techniques have been used and evaluated? Why is this particular case study of value to the community?

>>> We’ve added the text: “An experimental fire in a partially mulched black spruce stand (following an alternating strip pattern) showed minimal reduction in fire intensity due to the fire readily breaching the 4-6 m wide mulch gap between trees [4].  Fire behaviour in stands with mastication of only sub-canopy trees or shrubs has been documented [5]. Fire behaviour in completely mulched (with no stem removal or retention of standing trees) in low-volume boreal forest is unknown, despite the treatment being common along wide utility rights-of-way and at the edge of industrial sites in boreal western Canada.  “ to show the relevance of this particular fuel complex.

The site description can be improved by mentioning the soil type and soil moisture conditions. These are valuable information for a study about fuels.

>> We added the following to enhance the ecosystem and fuel description: “Soils underlying the mulch have an organic layer 10-15 cm at higher elevations, and 40 or more cm of organic soil at the north and western edges of the burn plot.  Clay mineral soils are found throughout the site.”

Further, a site map is needed that should include the location of the sites evaluated and the Sandy Lake tower. The site map could also be useful to show the burned area and extent, fuel composition and arragenement, and sensor locations.

>>> We added figures 1 and 2 to show the study site location, plot layout, and sensor arrangement.

Other comments:

L 11 where is Pelican Mountain research site?

>> we added details on the site location both in figure 1 as well as in the first sentence of section 2.1 the site description.

L12-13 this sentence is awkward, treated how? This is also a good place to say what was done in the study.

>> we have simplified the abstract’s first few sentences to: “An experimental fire was conducted in one year old mulched (masticated) boreal fuels. Infrared imagery was obtained to quantify rate of spread and intensity to a 1 m resolution. In-stand totalizing heat flux sensors allowed for the observation of energy release near the surface.”

L17 superscripts needed for units

>> done.

L18-19 specifically what is expected in a logging-slash fuel type?

>>> we added the text: “(dominated by fine branchwood and dead foliage)”to be specific to slash fuels in the Canadian context

L37-38 more information about the fire and area are needed in Section 2.1. Further, a site map is needed that should include the location of the sites evaluated.

>>> Figures 1 and 2 show details on the site location.

L55-60 should be introduced much earlier in the paper, such as in the introduction.

>> moved the text on the need for understanding fuel treatment fire behaviour to the end of the introduction

L59 what is Unit 5 -- what is the composition, general description of this site and why is is used for comparison?

>>> We’ve changed this text to: “Additionally we compare the results from this experimental fire with the results from an adjacent plot that was burned as an experimental crown fire in May 2019 [4].  “ which is a citation to a paper by Thompson et al from this same special issue.

L62 add the tower to a site map.

>>done

L64-65 the dates need a year

>> clarified this refers to 2017

Section 2.2 weather discussion would benefit from # of antecedent dry days. What is the average precipitation?

 >>> We now add the additional detail: “During the month of May in 2017, 20 of 31 days were below normal precipitation, though the monthly precipitation was average. From May 25 to May 31 no precipitation was recorded at the research site. In this absence of precipitation, the mulch fuels dried to a state that would permit combustion in the uppermost 5 cm layer of fuel [12].  “

Table 2 caption and table need to be improved. What is FWI, S-1, S-2, S-3, and M-2?

>> added the acronym titles in the table caption as suggested.

L99 define CFFDRS. Also need more description about the model that is used.

>> we added the following in the introduction: “The Canadian Forest Fire Danger Rating System (CFFDRS) contains two subsystems for defining wildfire hazard, the Canadian Forest Fire Weather Index System (FWI) and the Canadian Forest Fire Behaviour Prediction System (FBP). The FBP system is used to predict the potential spread and intensity of fire based on 17 standard fuel models. As fuels are modified, they no longer conform to the standard models defined. “

L102 needs a space between min-1(Figure)

>> done.

L105 quantify what “considerable fuel consumption” is

>> we clarified this is approximately 50% based on laboratory tests of flaming vs smouldering in mulch of the same size class distribution and origin.

Figure 2 – remove the titles at the top of the figure A and B (i.e. “rate of spread from point ignition in mulch fuel bed”) and incorporate into the Figure captions.

 >> Done.

L116 use a superscript for units in 1.8 kg/m2

>> done

L117-118 needs more context for those unfamiliar with this model. This should be described prior to the results (in the methods section).

>> We’ve now provided abstracted rate of spread values (fire spread rate relative to wind speed) for comparison.

L117-127 is “discussion” and should be moved

>>> we’ve extensively modified the discussion and results to better reflect observations vs discussion.

L124 spell out US

>>done

L131 what dates were the visits? What were the preceding and prevailing precipitation and temperature conditions like? How much shrub recovery, where, etc.?

>>> Revised to: “There was also notable shrub recovery within one year (May 2018) to approximately 50% cover, primarily of the ericaceous shrub Rhododendron groenlandicum

L150-152 show this or provide supporting evidence

>>> Since we did not quantify the decomposition rate, we’ve removed this observation.

L152-153, why is this important to note?

>>> This is important to note qualitatively as a recovery of a flammable surface fuel.  We’ve revised this sentence to note the flammability of the ericaceous shrubs.

L171 what is spotting

>> added the text “spotting (small fires ignited ahead of the main fire by lofted firebrands)” to clarify that definition.

L181 how can this case study be applied to other areas or management?

>>> we revise the conclusions to state that a mulch fire spread model for this fuel complex would be applicable across boreal North America.

Reviewer 3 Report

This manuscript assessed the fire behavior in a mulched black spruce through a designed experiment, aiming to better understand how mulching in low-volume boreal conifer stands affects fire behavior. The authors analyzed the rate of spread and fire intensity from ground sensors and helicopter-derived infrared images to compare the differences in terms of the pre-mulch fuel-type. Meanwhile, the authors also made a comparison between the experiment-derived results and the slash fuel model. Given the important role of fuel management techniques in conducting management activities and decision-making, this study is useful. However, for the current version, the structure of the manuscript is unclear (it looks like an experiment report), I did not see any value as an academic research paper to be published in ANY journal. My recommendation is a solid rejection. The manuscript needs to be carefully revised and re-wrote.

Major comments:

Introduction: What’s the research progress in literature? What’s the research gap that you have found? What are your research questions? I didn’t saw this important information which is critical to introducing your study.

Materials and Methods:

A figure of the study case, the spatial distribution of specific Units would help readers to understand your study.

Topography and wind direction/speed are critical for the fire spread and intensity, any contribution in your study?

L48-L59, does the mulch size class (SC1 (0-0.5 cm), SC2 (0.5 – 1 cm), SC3(1-3 cm) and SC4 (3-5 cm)) have impacts on your results?

L55-L60, here is the objective of this study, please move to the introduction section.

L91, where is Figure 3a?

L117-L127, these are the discussion, better to mover to the discussion section.

Author Response

We thank reviewer 3 for their comments and revise the manuscript accordingly as detailed in the responses below.

Reviewer 3

Open Review

English language and style

( ) Extensive editing of English language and style required
(x) Moderate English changes required
( ) English language and style are fine/minor spell check required
( ) I don't feel qualified to judge about the English language and style

Yes

Can be improved

Must be improved

Not applicable

Does the introduction provide sufficient background and include all relevant references?

( )

( )

( )

(x)

Is the research design appropriate?

( )

( )

(x)

( )

Are the methods adequately described?

( )

( )

(x)

( )

Are the results clearly presented?

( )

( )

(x)

( )

Are the conclusions supported by the results?

( )

( )

(x)

( )

Comments and Suggestions for Authors

This manuscript assessed the fire behavior in a mulched black spruce through a designed experiment, aiming to better understand how mulching in low-volume boreal conifer stands affects fire behavior. The authors analyzed the rate of spread and fire intensity from ground sensors and helicopter-derived infrared images to compare the differences in terms of the pre-mulch fuel-type. Meanwhile, the authors also made a comparison between the experiment-derived results and the slash fuel model. Given the important role of fuel management techniques in conducting management activities and decision-making, this study is useful. However, for the current version, the structure of the manuscript is unclear (it looks like an experiment report), I did not see any value as an academic research paper to be published in ANY journal. My recommendation is a solid rejection. The manuscript needs to be carefully revised and re-wrote.

Major comments:

Introduction: What’s the research progress in literature? What’s the research gap that you have found? What are your research questions? I didn’t saw this important information which is critical to introducing your study.

>>> we know more specifically point to the knowledge gap: “An experimental fire in a partially mulched black spruce stand (following an alternating strip pattern) showed minimal reduction in fire intensity due to the fire readily breaching the 4-6 m wide mulch gap between trees [4].  Fire behaviour in stands with mastication of only sub-canopy trees or shrubs has been documented [5]. Fire behaviour in completely mulched (with no stem removal or retention of standing trees) in low-volume boreal forest is unknown, despite the treatment being common along wide utility rights-of-way and at the edge of industrial sites in boreal western Canada.  “

Materials and Methods:

A figure of the study case, the spatial distribution of specific Units would help readers to understand your study.

>>> see revised Figure 2 for a detailed site layout.

Topography and wind direction/speed are critical for the fire spread and intensity, any contribution in your study?

>> we clearly state that the wind direction was constant and there is essentially no topography at the site.

L48-L59, does the mulch size class (SC1 (0-0.5 cm), SC2 (0.5 – 1 cm), SC3(1-3 cm) and SC4 (3-5 cm)) have impacts on your results?

>>> Yes, we discuss that now in the second paragraph of the discussion: “As a result, the fraction of very fine fuels (i.e. foliage) is much higher in slash than mulch, and is likely a contributor to the lower fire intensities observed here”.  The variations of mulch size between the classes is not the subject of the study here, but rather how it broadly contrasts with non-mulch fuels.

L55-L60, here is the objective of this study, please move to the introduction section.

>>done.

L91, where is Figure 3a?

>> we have revised all the figure numbering

L117-L127, these are the discussion, better to mover to the discussion section.

>> moved to the discussion.

Round 2

Reviewer 1 Report

There are still some minor sloppy edits in the paper and I have addressed some of those in the paper. Revisions look adequate.

Author Response

We thank the reviewer for their minor edits.  We've incorporated those and many other minor editorial revisions after our own extensive review as well.

Reviewer 2 Report

This study uses an experimental site and fire to better understand fire behavior in open mulch fuels. The authors focus on rate of spread and head fire intensity. They also evaluate the slash fuel model from the Canadian Fire Behavior Prediction System and compare to the experimental fire. This is an important topic, but the manuscript needs more detail before it should be published and there are still many minor inconsistencies and errors throughout the text. Specific suggestions to help improve the case study:

Results section – I suggest having two sub sections here. 3.1 can focus on the field observations and 3.2 should introduce the model and/or any other comparison data

Section 2.1 needs to describe the different Units presented in Figure 1. Also need to discuss what is the potential of Unit 5 to impact results in your study.

Abstract overall improved, but it is unclear the management implications of lines 18-24.

L29 be more clear, is this a global practice? Or specific to certain regions?

L33-34 remove “Click or tap here to enter text” and all other instances such as L80, L102, L107, L120

L53 generally what are the 17 standard fuels, it would be good to provide context on what these encompass.

L56-58 revise awkward sounding sentence

L58-59 sounds redundant with the start of the previous sentence

L63 what is the normal return interval? What is considered “short?”

L66-68 What is the slash fuel model?

L77 add a comma between the latitude and longitude

L81 add “the” between “of” and “mulched”

L87 revise grammar “is be”

L100-102 change repeated words such as “distribution” and “same”

L115 replace “for over 30 years” with the year the tower began recording data

L116-117 what is the average precipitation? What is considered below normal?

L136 revise awkward sounding sentence

L144 revise figure #s (these are off) as well as updating the order of figures referenced within text (if this # is correct, then it needs to be updated to figure 2 as no 2 and 3 have been introduced yet)

L151-152 revise reference format

L189 Table 3 is not discussed or referenced in the text

L194 Figure 1 is not correctly numbered

L198 Figure 4 is out of order

L204 check the table # referenced

L209-210 revise awkward sounding sentence

L252-254 recovery is mentioned, but doesn’t seem to fit at the end of this paragraph unless it is followed up with how it could potentially impact future fire patterns.

L259-260 the conclusion drawn here is still relatively weak. More discussion on this is needed.

Author Response

Results section – I suggest having two sub sections here. 3.1 can focus on the field observations and 3.2 should introduce the model and/or any other comparison data

 >>> Done.  We’ve split this into 3.1 Fire behaviour observations and 3.2 Comparison to fire behaviour models

Section 2.1 needs to describe the different Units presented in Figure 1. Also need to discuss what is the potential of Unit 5 to impact results in your study.

 >>> Added: “While a number of units are visible within Figure 1 the key units are Unit 1 and Unit 5, which were both lowland boreal spruce sites with mulching activity that were burned 1 year after treatment.”

Abstract overall improved, but it is unclear the management implications of lines 18-24.

 >>> Added “Typical mulching practices remove remnant biomass, leaving biomass in situ reduces overall management input.” In order to describe the reduction in rate of spread without typical management effort.

L29 be more clear, is this a global practice? Or specific to certain regions?

 >>> The practice is highly prevalent in Alberta, however it does occur abroad. Changed to: “As mulching is a prevalent fuels management technique in Alberta, Canada, future studies will contribute to the development of a fire behaviour prediction model”

L33-34 remove “Click or tap here to enter text” and all other instances such as L80, L102, L107, L120

 >>> Appears to have been an issue with the generation of the PDF. Has been addressed.

L53 generally what are the 17 standard fuels, it would be good to provide context on what these encompass.

 >>> To discuss in broad terms the fuels modelled within the FBP system, added “The 17 fuel models describe fire behaviour in five broad categories: C – coniferous fuels (7 models), D – deciduous fuels (1 model), M – mixedwood fuels (4 models), O – open (grass) fuels (2 models) and S – slash fuels (3 models).”

L56-58 revise awkward sounding sentence

 >>> Reformatted to: “Mulching is expected to reduce the overall rate of spread and intensity of fire in these fuels, however the magnitude of that reduction is difficult to assess without experimental case studies as this treatment does not remove fuels.”

L58-59 sounds redundant with the start of the previous sentence

 >>> Addressed above

L63 what is the normal return interval? What is considered “short?”

>> We’ve revised this sentence to emphasize that younger forests burning makes fuel treatments more important: “Nearby instances of rapid wildfire spread in recently-burned forest that has traditionally been considered a barrier to spread [7] increases the importance of a detailed understanding of fuel treatments if they are to be implemented near communities.”

L66-68 What is the slash fuel model?

 >>> Corrected to “slash fire behaviour model” as this refers to the FBP system slash fuel, fire behaviour model.

L77 add a comma between the latitude and longitude

 >>> Added

L81 add “the” between “of” and “mulched”

 >>> Added

L87 revise grammar “is be”

 >>> Revised to “is”

L100-102 change repeated words such as “distribution” and “same”

 >>> Amended to: “The mulch particle composition is assumed to follow the distribution measured in [9] at other mulched areas at the same site and equipment: SC1 58%, SC2 17%, SC3 20% and SC4 5%.”

L115 replace “for over 30 years” with the year the tower began recording data

 >>> Replaced with starting year “1977”

L116-117 what is the average precipitation? What is considered below normal?

 >>> Added average monthly precipitation “(47 mm)” and a qualifier for below normal “( < 1 mm daily)”

L136 revise awkward sounding sentence

 >>> Revised to: “Temperature and humidity far exceeded normal values of Sandy Lake’s 30-year noon fire weather observations temperature record (18°C) for May 31.”

L144 revise figure #s (these are off) as well as updating the order of figures referenced within text (if this # is correct, then it needs to be updated to figure 2 as no 2 and 3 have been introduced yet)

>>> Addressed figure numbering, figure 3 now properly identified and referenced in order.

L151-152 revise reference format

>> We are following the MDPI reference format as provided in the journal instructions to authors

L189 Table 3 is not discussed or referenced in the text

 >>> Discussed with reference to management capacity via the text: “Potential fire behaviour for a fire of this nature would have been assessed using a surrogate fuel model, in this case S-2. Notably, S-2 has a considerable head fire intensity of 28,000 kW m-1 yet the observed head fire intensity was 650 kW m-1 (Table 3).”

L194 Figure 1 is not correctly numbered

 >>> Now numbered as Figure 3

L198 Figure 4 is out of order

 >>> With Figure 3 in place this is now in the correct location

L204 check the table # referenced

 >>> Revised to table 3

L209-210 revise awkward sounding sentence

 >> Rephrased to “The observed experimental mulch fire was considerably less intense and slower spreading than any of the slash benchmark fire behaviour models (Table 3). Fire in mulch spread 88% slower than slash (S-2) and was 98% less intense. This demonstrates the need for additional data in mulch to define a unique fire spread model. Observations of fireline intensity declines due to mulching are broadly similar to masticated fuels in the U.S. [5], with notably thicker fuel beds, higher wind speeds, and consequently somewhat higher flame lengths in this study.”

L252-254 recovery is mentioned, but doesn’t seem to fit at the end of this paragraph unless it is followed up with how it could potentially impact future fire patterns.

 >>> We add the following sentence to the end of the paragraph: “Rapid post-fire shrub recovery in concert with remnant mulch and organic soil itself represents a novel fuel complex that is in need of further investigation, especially if grass species typically problematic after timber harvest in the region [23] emerge.”

L259-260 the conclusion drawn here is still relatively weak. More discussion on this is needed.

>>> Given that this is a case study, we left the conclusions relatively broad.  However, to your suggestion, we’ve modified the conclusion to now read: “This case study of fire behaviour in boreal mulched fuels has showed an effective reduction in spread rate and intensity during relatively high fire weather indices (hot and dry) down to levels within the realm of direct suppression.  High-resolution overhead infrared imagery revealed brief periods of fire intensity above levels of safe direct attack, despite mean intensity values well within safety limits for direct attack.  Experimental fires for assessing suppression capacity (e.g. [8]) should as much as possible employ overhead infrared progression mapping to capture both the mean and the variance in spread rates.  Slash fuel models were found to be inappropriate surrogates to boreal mulched fuels, and additional experiments are required to define a mulch spread model.  Future similar fires will be key in understanding the parameters under which mulch treatments are effective. Improved knowledge of the vegetation recovery and post-fire flammability will be key in understanding the efficacy of applying low-intensity prescribed fire to boreal mulch treatments.  This study and subsequent studies are intended to serve as data for the development of a mulch fuel type in black spruce dominated forests, which would be applicable across boreal North America.”

Where we now emphasize the methodological gains we saw from overhead IR imagery in terms of quantifying variability around the mean rate of spread and intensity.

Round 3

Reviewer 1 Report

Thank you for clarifying my concerns. I believe now it is a much improved paper.

Author Response

We thank the reviewer for their efforts during this revision cycle of this paper.